# Sparse Skill Coding: Learning Behavioral Hierarchies with Sparse Codes

## Abstract

Many approaches to hierarchical reinforcement learning aim to identify sub-goal structure in tasks. We consider an alternative perspective based on identifying behavioral 'motifs'—repeated action sequences that can be compressed to yield a compact code of action trajectories. We present a method for iteratively compressing action trajectories to learn nested behavioral hierarchies of arbitrary depth, with actions of arbitrary length. The learned temporally extended actions provide new action primitives that can participate in deeper hierarchies as the agent learns. We demonstrate the relevance of this approach for tasks with non-trivial hierarchical structure and show that the approach can be used to accelerate learning in recursively more complex tasks through transfer.

## 1 Introduction

Despite the many successes of deep reinforcement learning (RL) in recent years (Mnih et al., 2015; Schulman et al., 2017; Silver et al., 2016; Levine et al., 2016), long-term credit assignment and search complexity remain fundamental challenges. One of the primary strategies for managing this complexity has been to incorporate hierarchical, temporally-extended actions. Hierarchies hand-designed using domain knowledge can provide substantial training benefits (Sutton et al., 1999; Barto & Mahadevan, 2003). However, a major challenge in hierarchical reinforcement learning is to develop general methods for discovering useful hierarchical representations without relying on domain expertise.

Many objectives for the hierarchy learning problem have been proposed, with notable focus on facilitating transfer to downstream tasks and facilitating efficient exploration of the state space (Eysenbach et al., 2018; Frans et al., 2017; Solway et al., 2014). We pose the hierarchy learning problem as follows: given a distribution of tasks, what determines the optimal set of representations for action sequences? We approach this question by considering the problem faced by human decision-makers. Humans are fundamentally resource-constrained. Energy is limited, computation is expensive, and solutions to problems must be computed in real-time. Across cortical areas, a common strategy for dealing with these constraints is to reduce computational complexity by storing representations that *efficiently encode* the statistics of the domain. This idea originated as the efficient coding hypothesis (Barlow, 1961), and has been empirically corroborated in sensory and motor systems (Barlow, 1961; Olshausen & Field, 1996; Hromádka et al., 2008; Poo & Isaacson, 2009; Vinje & Gallant, 2000).

We extend the efficient coding hypothesis to the problem of representation learning for planning, and propose a method for discovering temporally extended actions by learning an efficient code of the behavior required by a task distribution. This is a novel formulation of the classic notion of "chunking" from cognitive psychology (Chase & Simon, 1973; Simon, 1991), which motivated early work on hierarchical reinforcement learning (Korf, 1985; Stolle & Precup, 2002), and aligns with empirical neuroscience results suggesting that organisms represent their motor output in terms of a sparse efficient code of high-level "motor primitives" (Flash & Hochner, 2005). An efficient code for a sequence of actions compresses the sequence into a minimum-length description that factorizes the input distribution (Cover & Thomas, 2012). The benefit of using such a code in the context of decision making is that it provides the building blocks for solving related problems using a minimal set of decision points, delineated by a minimal set of skills that capture the statistics of the behavior required by the task distribution. As a consequence, this approach subsumes several distinct

objectives for hierarchy learning proposed in the literature; an efficient code of behavior required by a problem space reduces the number of decision points required (Harb et al., 2017), facilitates transfer to tasks drawn from the same distribution (Solway et al., 2014), facilitates efficient exploration of the state space [cite], and decomposes a task into a natural set of sub-tasks (Bacon et al., 2017; Fox et al., 2017).

The problem of finding a compact code for sequential data with long-range dependencies and nested hierarchical structure is equivalent to the problem of finding a minimum-length program that can generate the data—that is, finding a program with minimum Kolmogorov complexity (Kolmogorov, 1965). The Kolmogorov complexity of a sequence is not finitely computable and can thus only be approximated. Drawing inspiration from this idea, we propose a relatively simple approach for approximating minimum-description-length codes of sequences of actions through iterative convolutional sparse coding and compression, with structure similar to classic string compression methods such as the Nevill-Manning algorithm (Nevill-Manning & Witten, 1997). With this method, we are able to extract compact, hierarchically nested representations of action trajectories, with temporally extended actions of arbitrary lengths. We incorporate this method into the RL problem by equipping the agent with the capacity to compress its behavior and augment its action space with the learned representations after each task it faces.

## 2 PRELIMINARIES

**Reinforcement learning:** We consider a *task* as a finite horizon Markov decision process, consisting of a set of states $S$, a set of actions $A$, a transition function $p\left(S_{t+1} = s' | S_t = s, A_t = a\right)$ that defines how actions move an agent between states, and a reward function $r\left(S_t = s, A_t = a, S_{t+1} = s_{t+1}\right)$ that defines the reward the agent receives by taking action $a$ in state $s$ and ending up in state $s_{t+1}$. The objective is to find the policy $\pi : S \rightarrow A$ the expected cumulative reward

$$\mathbb{E}_\pi \left[ \sum_{t=1}^{T} r\left(S_t = s, A_t = a, S_{t+1} = s_{t+1}\right) \right].$$

**Sparse coding:** Sparse coding (Olshausen & Field, 1996) is an unsupervised algorithm for learning a dictionary that reconstructs a signal using a minimal set of non-zero coefficients on the dictionary. The sparse coding model assumes that a signal $X$ is generated as a linear combination of filters $W$ with coefficients $s$ plus additive Gaussian noise $\epsilon$:

$$x_i = \sum_{j=1}^{n} W_j s_{i,j} + \epsilon \tag{1}$$

The objective is to reconstruct the input with minimal distortion while using the minimal number of non-zero coefficients. The filters and coefficients are jointly optimized:

$$\min_{w,s} \sum_{i=1}^{m} ||x_i - \sum_{j=1}^{n} W_j s_{i,j}||_2^2 + \lambda \sum_{i,j} ||s_{i,j}||_1 \tag{2}$$

with the $l1$ norm imposing a penalty on the number on non-zero activations, and $\lambda$ modulating the trade-off between the accuracy and sparsity of the representations. To learn sparse codes for time series data, one can augment the basic sparse coding model by replacing scalar-valued coefficients with vector-valued coefficients and matrix multiplication by convolution, which allows basis functions to appear at all possible shifts in the signal:

$$\min_{w,s} \sum_{i=1}^{m} ||x_i - \sum_{j=1}^{n} W_j * s_{i,j}||_2^2 + \lambda \sum_{i,j} ||s_{i,j}||_1 \tag{3}$$

We use this convolutional formulation of the sparse coding model to encode trajectories of actions generated by an RL agent. We note the similarity between the sparsity constraint and the Minimum Description Length (MDL) principle, which states that the best model $\hat{M} \in \mathcal{M}$ is that which can describe a data sample $x$ completely using the fewest number of bits,

$$\hat{M} = \underset{M \in \hat{M}}{\arg \min} \, L(x, M)$$

where $L(x, M)$ is the codelength assignment function defining the theoretical code length required to describe $(x, M)$ uniquely. Underlying the MDL is the idea that a model that is able to (losslessly) compress data must do so by capturing its structure and regularities. We use sparse coding to approximate this objective.

## 3 SPARSE SKILL CODING

We propose a method, sparse skill coding (SSC), for discovering hierarchically nested codes for action sequences using a variant of convolutional sparse coding. Given a trajectory $\tau \in \mathbb{Z}^t$ consisting of $t$ timesteps of $n$ discrete actions, we wish to find a minimal set of multi-step actions that encodes this trajectory. We represent trajectories as binary matrices $T^{n \times t} \in [0, 1]$, where actions are one-hot encoded.

The standard sparse coding model learns a single layer code and requires fixing the size of the dictionary elements (the length of the actions) in advance. We propose an alternative method that can discover potentially hierarchically-nested dictionary elements of arbitrary length with an iterative coding and compression process.

At all stages, the size of the dictionary elements is set to 2-timesteps. A dictionary and sparse code is found for the batch of trajectories, by minimizing equation 2. The dictionary element $a$ which has the highest explained variance is then selected and assigned an integer code $n + 1$. The dimension of the matrix $T$ is increased to $T^{n+1 \times t}$. All 2-step time windows that yielded an active coefficient on this dictionary element are then replaced with a 1-step one hot vector encoding the dictionary element's integer code $n + 1$. The length of the trajectory is thus decreased by the number of occurrences of that dictionary element $w$. We denote this compression procedure with the function $\Gamma(T, a)$. This process is repeated for the new matrix $T^{n+1 \times t-w}$, for $k$ iterations. In this manner, dictionary elements can be discovered that contain previously compressed sequences.

---

**Algorithm 1:** Sparse Skill Coding

**Input:** Batch of $m$ trajectories encoded as binary matrices $T^{n \times t} \in [0, 1]$
**Output:** Dictionary of $K$ high-level actions $D_K$

1 **for** $k = 1$ *to* $K$ **do**
2      $\min_{w,s} \sum_{i=1}^{m} ||T_i - \sum_{j=1}^{n} W_j * s_{i,j}||_2^2 + \lambda \sum_{i,j} ||s_{i,j}||_1$
3      $s_\omega = \arg \max \sum_i^m s$
4      $D_k = W_\omega$
5      $T_k = \Gamma(T, D_k)$
6 **endfor**
7 **return** $D_K$

---

The result of this process is a set of (potentially nested) high-level actions that capture the statistical structure of trajectories generated on a task. An agent's action space can then be augmented to include these high-level actions, which can facilitate transfer to tasks drawn from the same distribution.

## 4 RELATED WORK

Early work in hierarchical reinforcement learning demonstrated that well-designed sub-goals or high-level actions can significantly speed the discovery of shortest-path solutions (Sutton et al., 1999; Barto & Mahadevan, 2003; Dayan & Hinton, 1993) and facilitate transfer to related tasks (Konidaris & Barto, 2007). Later work demonstrated the advantages of incorporating pre-defined sub-goals into deep reinforcement learning (Kulkarni et al., 2016) or pre-learned skills (Tessler et al., 2017), but left open the question of how to discover these sub-goals or skills automatically.

Recent work have attempted to discover these temporally extended actions by optimizing for reusable behaviors shared across tasks (Frans et al., 2017), maximizing diversity in exploration (Florensa et al.,

2017; Eysenbach et al., 2018; Gregor et al., 2016; Achiam et al., 2018), or by finding bottlenecks in demonstrations (Kipf et al., 2018; Co-Reyes et al., 2018), after which these temporally extended actions are combined with a high-level policy to learn on downstream tasks.

However, in contexts in which bottleneck states are less apparent, approaches for end-to-end learning of temporally extended actions and policies, such as options (Bacon et al., 2017; Harb et al., 2017) frequently degenerate to learning either single-step options or only a single option for the entire trajectory. On the other hand, approaches that mitigate this degeneracy by fixing the horizon length of each sub-policy (Nachum et al., 2018; Frans et al., 2017). Furthermore, while in theory methods such as options (Sutton et al., 1999) or hierarchies of abstract machines (Parr & Russell, 1998) could learn nested behavior, in practice because the number of contexts grows exponentially with depth, most approaches focus on learning two-level hierarchies, with the exception of (Fox et al., 2017) which proposes a method for learning deeper nested hierarchies, but with a fixed number of options available at each depth.

Nested structure is characteristic of problems in natural language processing (Socher et al., 2011) or program induction (Parisotto et al., 2016), but approaches in these fields usually have access to additional top-down supervision on tree structure. Our method discovers variable-length temporally extended actions in a bottom-up fashion from demonstration , and we show that our method is able to nest temporally extended actions and transfer to recursively structured environments where bottlenecks are not that clearly apparent.

## 5 EXPERIMENTS AND RESULTS

In our experiments, we ask the following questions:

- Can sparse skill coding learn temporally extended actions that reflect the nested hierarchy of a task?
- Do the temporally extended actions learned from sparse skill coding better capture *behavioral motifs* than hierarchical RL methods that learn to identify sub-goals?
- Can an agent transfer these temporally extended actions to learn more quickly on a series of recursively more complex environments?

We find that in contrast to those learned in subgoal-based hierarchical approaches, the temporally extended actions learned from sparse skill coding reflect commonly repeated patterns of behavior that can be used to build a nested hierarchy, and such a nested hierarchy enables the agent to continually transfer to recursively more complex environments.

To evaluate our approach, we consider the Lightbot domain (explained in further detail in Section 5.1.1) and the classic four rooms domain. We compare with an option-critic baseline (Bacon et al., 2017) trained with proximal policy optimization (Schulman et al., 2017).

### 5.1 EXPERIMENT 1: LEARNING SPARSE SKILLS FROM DEMONSTRATION

To understand the properties of representations learned with this method, we first present a qualitative analysis of the representations learned by sparse skill coding performed on trajectories generated by an expert policy, and contrast these learned representations with those learned via option-critic (Bacon et al., 2017) on the same task.

#### 5.1.1 TASKS

We compare the abstractions learned by sparse skill coding and option-critic on a task that highlights the relevance of identifying behavioral "motifs" over subgoal states.

**Lightbot:** The Lightbot domain (Figure 1) is adapted from a game developed to teach children how to program. For each level in the game, there exists a compact, hierarchical program that generates the solution. In the original game, the objective is to find the shortest program that solves the level. Whereas Sanborn et al. (2018) used the Lightbot domain to study hierarchical learning in humans, we adapt the Lightbot game as a novel domain for hierarchical RL methods: the agent begins in a random location and direction in the room and must navigate the room to turn on all of the lights

(blue tiles) using five basic actions: `walk`, `jump`, `right`, `left`, and `light` (which turns on the light). This domain presents a challenging sparse reward task: the agent receives a positive reward of only if it successfully turns off all lights.

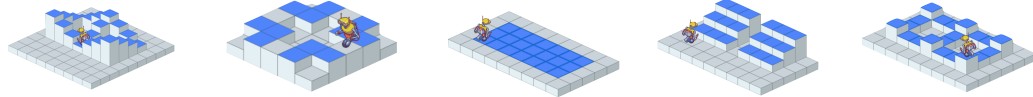

Figure 1: The Lightbot domain.

### 5.1.2 SUB-GOALS VS. MOTIFS

The repeated patterns in the solutions for Lightbot puzzles serve to test whether methods that discover nested hierarchical structure, such as ours, are able to learn re-usable temporally extended-actions that better reflect the structure of the environment than methods that chain together subtrajectories between sub-goals. Figure 2 visualizes the action sequences generated while optimizing a policy with proximal policy optimization (PPO) (Schulman et al., 2017) in the Lightbot and Four Rooms domain. In environments with nested hierarchical structure, such as the Lightbot domain, compressible sequential structure emerges in the agent's action sequences. This structure can be compactly encoded with a short, hierarchical code. In domains more conducive to sub-goal approaches, such as Four Rooms, behavioral motifs are less apparent; solutions chain together sequences of repeated actions (e.g. [`right`, `right`, `right`], [`down`, `down`, `down`]). Such sequences could be compressed with a run-length encoding scheme, but lack the nested structure that requires hierarchical compression schemes.

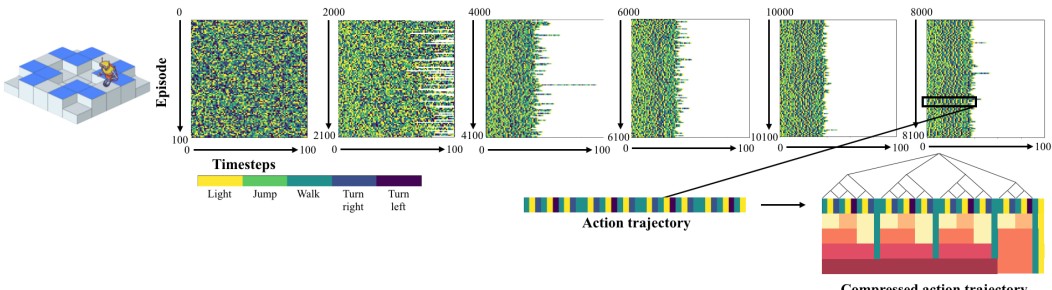

Figure 2: Hierarchical structure in the trajectories of a PPO agent in the Lightbot domain.

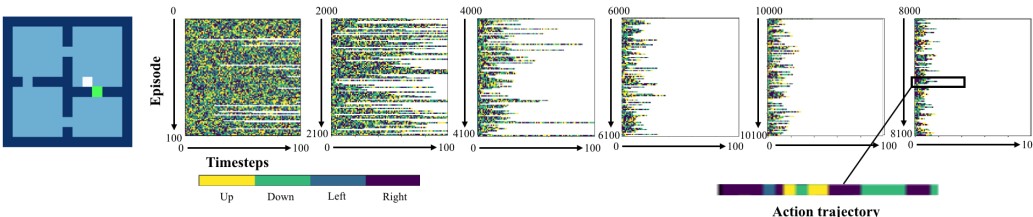

Figure 3: Convergence of action trajectories in the four rooms domain. Converged trajectories do not contain the hierarchically nested structure present in the Lightbot domain.

### 5.1.3 RESULTS

An expert policy was obtained with PPO on the Lightbot puzzle in Figure 4 under a shaped reward structure, where +10 reward was received for every light turned on, and -1 for every other action. The policy was trained to convergence with a learning rate of $10^{-5}$ for 10,000 episodes, with a gradient update every 100 episodes and a maximum episode length of 100 timesteps. A batch of

1,000 trajectories was generated from the converged policy and encoded with sparse skill coding for 8 iterations, yielding a set of 8 nested hierarchical actions. The action space of a new agent was augmented with these 8 actions and its policy was trained to convergence.

We compare the skills learned with sparse skill coding to options learned with option-critic (Bacon et al., 2017) trained with PPO. The same hyperparameters were used for both algorithms, with the addition of a deliberation cost (Harb et al., 2017) of $0.05$ for option-critic.

Figure 4 shows the normalized cumulative terminations per state for each option and skill learned by these two methods in the Lightbot domain. Options learned with option-critic show some specialization, but are highly redundant and fail to capture the nested structure inherent in the task. Sparse skill coding learns separable skills that reflect the structure of the environment.

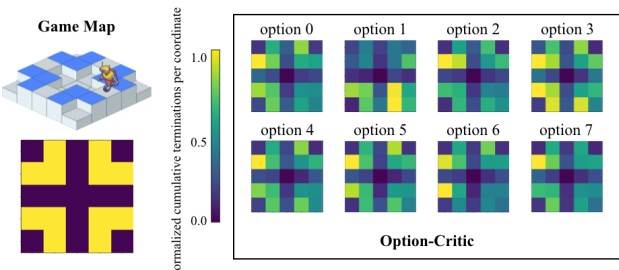 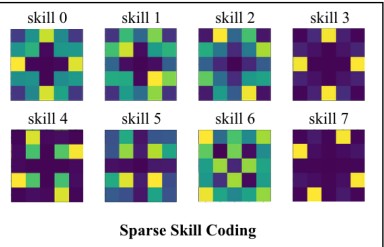

Figure 4: Normalized cumulative terminations per state for options learned via option-critic (left) and skills learned via sparse skill coding (right).

## 5.2 EXPERIMENT 2: LEARNING SKILLS FOR CONTINUAL TRANSFER

A motivation for learning high-level temporally-extended actions in the first place is that it reduces the *cognitive cost* of choosing a series of actions to the cognitive cost of choosing only one action. Therefore, the potential benefit of discovering behavioral motifs as high-level actions is that such high-level actions not only could be re-used in various related contexts, but could serve as primitives for building even higher-level motifs for even more complex domains. The decision to add a high-level action to the agent's repertoire of skills pays upfront the cognitive cost of taking that particular series of primitive actions, such that the agent need not pay such a cost when invoking the high-level action for future learning.

We are interested in understanding the implications that the iterative encapsulation of higher and higher-level actions have as the agent faces a task more complex than tasks it has trained on previously. Environments that exhibit a recursive or fractal structure, such as the Tower of Hanoi, offer a natural suite of tasks that grow rapidly in complexity from the perspective taking primitive actions, but whose solutions are straightforward if sub-solutions to easier problems may be re-used. Many real-world problems have many nested layers of complexity and such fractal environments boils such nested structure into its purest form, allowing us to take a first step towards understanding how an intelligent agent may re-use primitive sub-solutions to enable learning on more complex versions of problems it has encountered.

In the quantitative results that follow, we are not as interested in the asymptotic performance of SSC compared to standard approaches for continual transfer as much as we are in the speed at which SSC adapts as well as the compositional structure of the trajectories that SSC learns.

### 5.2.1 TASKS

We consider two domains, the Tower of Hanoi puzzle (Figure 5) and Fractal Lightbot (Figure 6).

These domains are organized into two levels each as follows:

| Level | Tower of Hanoi | Fractal Lightbot |
|-------|----------------|------------------|
| 0 | 2 disks | 1 cross |
| 1 | 3 disks | 2 crosses |

For level 1, we initialize an SSC agent with weights from a PPO agent trained on level 0 and with an augmented action space created from encoding trajectories from the PPO agent trained on level 0. We also compare with PPO and option-critic agents that were (1) trained from scratch and (2) transferred from the previous level 0.

**Tower of Hanoi:** The Tower of Hanoi is a classic puzzle that has been extensively studied in cognitive psychology and planning (Anderson, 1990). In the Tower of Hanoi, the player must move a stack of $n$ disks from one peg to another by moving each disk one at a time, with the restriction that the player cannot place a larger disk on top of a smaller one. One of the notable properties of this task is that the graph of its state space is a fractal resembling the Sierpinski triangle. Due to its cyclic nature, the optimal solution to the task is a recursive algorithm, which requires $2^{n-1}$ moves. We note that the recursive structure of this tasks can be exploited by an agent transferring its learning across tasks of increasing complexity, as the solutions to the $n$ step problem are contained within the solutions to the *n+1* step problem.

We model each Tower of Hanoi puzzle as a sparse reward reinforcement learning problem in which the agent receives a reward of 0 from the environment for every action taken and a reward of 10 for successfully transferring the tower of disks. In addition, the agent incurs a cognitive cost of -1 for every action taken. On each episode, the tower of disks is initialized on a random peg.

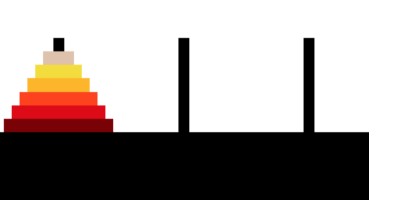 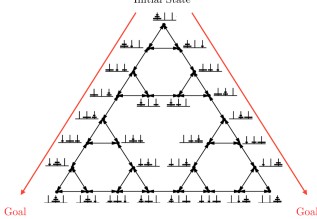

Figure 5: (Left) The Tower of Hanoi. (Right) State space for the three disk problem.

**Fractal Lightbot:** Fractal Lightbot is an adaptation of the Lightbot puzzles built on top of the Minigrid environment (Chevalier-Boisvert et al., 2018), which permits the use of images as state representations. The dimensions of the observations are fixed as the complexity of the puzzles increases; the agent's observations are 9x9 images showing overhead views of the portion of the environment that is directly in front of the agent, which changes as the agent moves around. Unlike the original Lightbot game, we removed the possibility of tiles at multiple heights, and the agent is able to turn on the light when it is in front of rather than on top of the light tile so that it is not occluded.

We also model each Fractal Lightbot puzzle as a sparse reward reinforcement learning problem in which the agent receives a reward of 0 from the environment for every action taken and a reward of 100 for successfully transferring the tower of disks. In addition, the agent incurs a cognitive cost of -1 for every action taken. On each episode, the agent is initialized in a random location and direction.

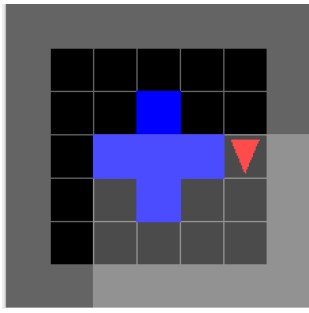 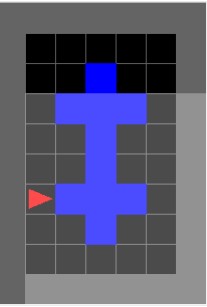

Figure 6: Lightbot puzzles that grow in a fractal manner. *left*: level 0, *right*: level 1

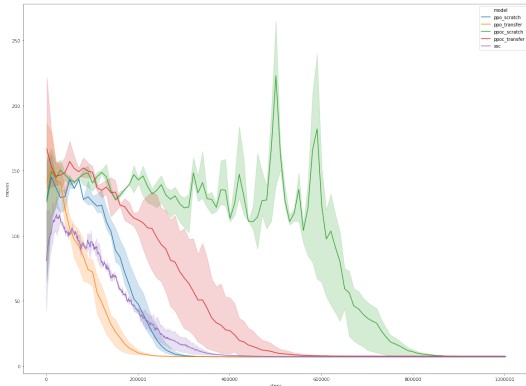

Figure 7: Transfer learning in the Tower of Hanoi. Each line averages over 3 different random seeds. Error bars show 95% confidence intervals.

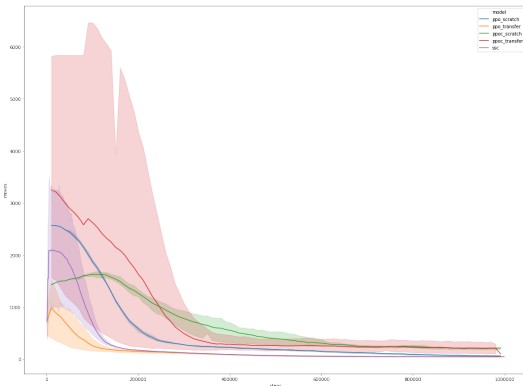

Figure 8: Transfer learning in Fractal Lightbot. Each line averages over 3 different random seeds. Error bars show 95% confidence intervals.

### 5.2.2 RESULTS

Figure 7 compares SSC with our baselines on transferring from level 0 (2 disks) to level 1 (3 disks) for Tower of Hanoi, and Figure 8 compares SSC with our baselines on transferring from level 0 (1 cross) to level 1 (2 crosses) for Fractal Lightbot. We observe that SSC performs much better than option critic. SSC transfer slightly slower than PPO, possibly because exploring with long high-level actions potentially is more costly.

## 6 DISCUSSION

Our goal is (1) to understand how to design an algorithm that can discover nested behavioral hierarchies of arbitrary depth, with actions of arbitrary length and (2) understand how reducing the cognitive cost of choosing actions with high-level actions affect transfer. Sparse skill coding is our method for studying these questions. As our method is a bottom-up method for learning nested hierarchies of temporally extended actions, we are able to avoid the computational complexity that make learning nested hierarchies of more than two levels difficult. We have also shown the distinction between hierarchies characterized by subgoals and hierarchies characterized by recurring motifs. We hope this paper motivates future work in learning nested behavioral hierarchies.

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

# A DETAILS FOR EXPERIMENT 2

## A.1 AGENT DETAILS

**Tower of Hanoi:** Observations for an $m$-disk, 3-peg Hanoi task are represented with $m \times 3$ vectors encoding the location of each disk. Each action is parameterized as `source_peg, target_peg` which automatically moves the topmost disk on the `source_peg` on top of the topmost disk of the `target_peg`. Observations are encoded with a 3-layer fully-connected network with a hidden dimension of 256 units and ReLU activations at each layer. The last layer produces the action distribution.

**Fractal Lightbot:** Observations for the Fractal Lightbot task are encoded with a 3-layer CNN with hidden dimensions of 16, 32, and 64 to encode the image observations, with kernels of size $(2, 2)$, stride of 1, and 2 fully-connected output layers of 256 dimensions, with ReLU activations at every layer. The last layer produces the action distribution.

## A.2 TRAINING DETAILS

For level 1, we initialize an SSC agent with weights from a PPO agent trained on level 0 and with an augmented action space created from encoding trajectories from the PPO agent trained on level 0. We compare with the following baselines:

- A PPO agent trained on level 1 from scratch.
- A PPO agent trained on level 1 with weights initialized from training a PPO agent on level 0.
- An option-critic agent trained on level 1 from scratch.
- An option-critic agent trained on level 1 with weights initialized from training an option-critic agent on level 0.

SSC is trained using PPO. Because the environments are all sparse reward environments, we collect the minimum amount of whole episodes whose aggregate number of transitions is greater than or equal to 4096 before doing every gradient update. For PPO we use a clip ratio of 0.1 and a weight decay penalty of 1e-5. Option critic was initialized with 4 options.

The hyperparameters for each agent to converge were found using an informal search:

- For Fractal Lightbot, we used a learning rate decay of 0.99 every 100 updates. For Tower of Hanoi, we used a learning rate decay of 0.95 every 100 updates.
- We did not set a fixed horizon for the episodes and trained all models for 1,000,000 transitions. For training PPO and option critic from scratch on level 1 in Fractal Lightbot, we trained for 3,000,000 transitions because this was the amount needed for the agents to converge.
- When transferring from level 0 to level 1, we found that initializing the agents from the last checkpoint from level 0 had a difficult time exploring the new level because (1) the environment has sparse rewards and (2) the weights were optimized for level 0 only. Therefore, we initialize all agents from a checkpoint saved one-third the way through training before the return curve plateaus. For every task this point occurred after the agent has trained on about 4000 episodes, so we used the checkpoint at 4000 episodes as a standard.

