# OpenReview forum: "Sparse Skill Coding: Learning Behavioral Hierarchies with Sparse Codes"
_ICLR.cc/2020/Conference — Reject_

### Official Review · AnonReviewer3 · 2019-10-23
**Official Blind Review #3**

**Rating:** 3

**Review:**

This paper aims to propose a new way of discovering a specific type of temporal abstraction, the pattern of actions. To achieve this goal, it applies the sparse coding method to discover an efficient encoding of the action sequences generated by the agent's interaction with the environment. The filters/dictionaries discovered by the method then represent certain patterns of actions.

In general, I think it proposes an interesting view of the temporal abstraction. Although this kind of temporal abstraction is only valid in the certain types of environment (For example, as it only represents certain patterns of action sequence, it suffers from non-optimality in stochastic environment where no fixed action sequence would be optimal for solving the problem), the paper, especially the experiment part, clearly tells its readers in what situation can we expect it to perform well. In this sense, I think it provides some scientific insights that benefit my understanding.

However, there are many places in the paper making me confused. Therefore I can not fully understand the paper and can not accept it. My concerns are:

1. Equations 1, 2, and 3 are loosey-goosey. For example, there is no definition of x_i, is it a vector or a scalar? Similarly, there is no definition of W and s.
2. paragraph 3 in section 3 is extremely hard to understand. For example, the first sentence: "At all stages, the size of the dictionary elements is set to 2-timesteps". Where does the "2-timesteps" come? I have no idea what it is talking about.
3. in the algorithm, T_i is not defined.
4. in the experiment part, figure 4 only provides the termination of options/skills but doesn't provide corresponding policies/action sequence, which makes me hard to evaluate the result.
5. figure 7 and 8 seems to be contradicted with the paper's claim. And the author didn't give a reasonable explanation.


**Experience Assessment:**

I have read many papers in this area.

**Review Assessment: Checking Correctness Of Derivations And Theory:**

I assessed the sensibility of the derivations and theory.

**Review Assessment: Checking Correctness Of Experiments:**

I assessed the sensibility of the experiments.

**Review Assessment: Thoroughness In Paper Reading:**

I read the paper at least twice and used my best judgement in assessing the paper.

---

### Official Review · AnonReviewer1 · 2019-10-23
**Official Blind Review #1**

**Rating:** 1

**Review:**


The paper proposes a method that aims at encoding trajectories (described as a sequence of actions) into a set of discrete codes with a hierarchical structure. The principle of the algorithm (as far as the article allows me to understand) is to apply multiple iterations of classical sparse coding over the trajectories. The experimental section on simple (deterministic) tasks shows that the SSC method is able to extract interesting options, which can then be used to learn faster on some close domains.

In terms of positioning, I find the idea of the paper interesting (i.e encoding trajectories through discrete symbols) since it uses sparse coding approaches which, as far as I know, are not classical in the RL domain. This type of approach can give us both a meaningful insight about the "nature" of the learned policy (as it is the case in the paper that compresses expert trajectories), and can also become a manner to constraint an RL algorithm to force it to exhibit behaviors that could seem more natural to humans.

But the way it is done in this article is disappointing. First of all, the article is badly written, and I am still not sure to fully understand how the algorithm exactly works. Indeed, many notations are not well defined (see at the end of the review), and it makes the algorithm 1 difficult to catch. Then, the authors consider that trajectories are represented as sequences of actions (using one-hot encoding) and do not discuss this hard choice: representing trajectories as a sequence of actions usually rely on the assumption that both the environment is deterministic, and the initial state is always the same. Is it the case in this paper? If it is, it clearly restricts the applicability of the technique. If it is not, then I don't see how it could work well... As far as I understand, all experimental environments are deterministic. So the algorithm description would clearly need to be rewritten, and the authors have to discuss the assumptions they are doing mainly: deterministic environments and also the fact that the "options" can only be extracted once a first policy has be learned (or by using expert traces) which limits its applicability.

In terms of experiments, the assumption made is that we have access to a set of 'good' trajectories (which is easy in the proposed environments, but may be difficult in the real-life). It is compared to the option-critic architecture which simultaneously learns the options and the policy and I think that the comparison is somehow unfair. Since SSC is more a "sequence compression" algorithm, I would prefer to compare with existing sequence compression algorithms like hierarchical recurrent neural networks for instance.  The results are illustrated in very simple environments and the article would gain by using more complex ones (for instance the Atari grand challenge dataset could be used for such a study). So it is difficult to understand if the approach as it is is really interesting and efficient for general RL purposes.

Summary: A good idea, but not well described, with strong assumptions not discussed, and with low-quality experimental results.

Some other minor remarks:
The introduction is a little bit messy and does not well allow one to understand the focus on the paper, mixing some notions of neuroscience with classical reinforcement learning aspects, the connection between the two domains being not trivial.

Equation 2 versus Equation 3: What is the difference?
s notation appears in 2.1 and 2.2 while it corresponds to different things. The variables are not defined and we don't know in which domain they rely on.
Articulation between sparse coding and MDL not clear (since sparse coding is directly a way to minimize the MDL). MDL never used after that.
section 3, paragraph 3: I do not understand what is described here. The description has to be rewritten to allow the readers to understand the algorithm e.g "the size of the dictionary elements is set to 2-timesteps. " ?  "The dictionary element a which has the highest explained variance is then selected and assigned an integer code n + 1 " Variance on what ?  what is T_i ?
[cite] appears in the introduction


**Experience Assessment:**

I have published in this field for several years.

**Review Assessment: Checking Correctness Of Derivations And Theory:**

N/A

**Review Assessment: Checking Correctness Of Experiments:**

I assessed the sensibility of the experiments.

**Review Assessment: Thoroughness In Paper Reading:**

I read the paper thoroughly.

---

### Official Review · AnonReviewer2 · 2019-10-23
**Official Blind Review #2**

**Rating:** 6

**Review:**

The paper discusses identifying motifs for aiding in the solving of cognitive tasks when using Reinforcement Learning. The idea seems quite novel, but the presentation seems to be more complicated than it needs to be for the idea. For example the introduction is quite hard to parse and when you get down to it, the ideas don’t seem that complex.

The discussion of the technique seems to be lacking in detail. I would be hard-pushed to reproduce the work from the material presented.

Figure 2 is complex and lacks enough discussion in the text.

Figure 3 is likewise complex and is not mentioned at all in the text.

Figure 4 needs more discussion.

The results presented are quite minimal and don’t fully explore and evaluate the approach taken.

Specific issues:
- Page 2: broken citation: “state space [cite], “

- “Lightbot: The Lightbot domain … a positive reward of only if it successfully turns off all lights.” - this seems to be the opposite of all previous statements which talked about Turing lights on.

- “We also model each Fractal Lightbot puzzle … and a reward of 100 for successfully transferring the tower of disks.” - this sounds more like the reward for the tower.

**Experience Assessment:**

I have read many papers in this area.

**Review Assessment: Checking Correctness Of Derivations And Theory:**

I assessed the sensibility of the derivations and theory.

**Review Assessment: Checking Correctness Of Experiments:**

I assessed the sensibility of the experiments.

**Review Assessment: Thoroughness In Paper Reading:**

I read the paper thoroughly.

---

### Author Response · Authors · 2019-11-15
**Thank you for the thoughtful feedback and suggestions**

We appreciate the thoughtful feedback on our paper from all 3 reviewers. We believe that substantial revisions to the exposition and experiments are required to properly communicate and demonstrate our approach. This includes 1) comparisons to alternative sequence compression based approaches, 2) a clearer analysis of macro actions learned with our method and others, 3) experiments that more clearly demonstrate the extent of our approach’s generality and limitations, and 4) substantial changes to the exposition. We plan to address these concerns in a future version that we will submit elsewhere. Thank you to the reviewers for your time and careful reading.

---

### Decision · Program_Chairs · 2019-12-19

**Decision:**

Reject

**Comment:**

The paper proposes an interesting idea of identifying repeated action sequences, or behavioral motifs, in the context of hierarchical reinforcement learning, using sparsity/compression.  While this is a fresh and useful idea, it appears that the paper requires more work, both in terms of presentation/clarity and in terms of stronger empirical results.